# Loss of Heterozygosity in the Tumor DNA of De Novo Diagnosed Patients Is Associated with Poor Outcome for B-ALL but Not for T-ALL

**DOI:** 10.3390/genes13030398

**Published:** 2022-02-23

**Authors:** Natalya Risinskaya, Yana Kozhevnikova, Olga Gavrilina, Julia Chabaeva, Ekaterina Kotova, Anna Yushkova, Galina Isinova, Ksenija Zarubina, Tatiana Obukhova, Sergey Kulikov, Hunan Julhakyan, Andrey Sudarikov, Elena Parovichnikova

**Affiliations:** 1National Research Center for Hematology, Novy Zykovski Lane, 4a, 125167 Moscow, Russia; risinska@gmail.com (N.R.); dr.gavrilina@mail.ru (O.G.); uchabaeva@gmail.com (J.C.); 2017e.s.kotova@gmail.com (E.K.); ann.unikova@bk.ru (A.Y.); rara4v1@gmail.com (G.I.); ksenijazarubina@mail.ru (K.Z.); obukhova_t@mail.ru (T.O.); smkulikov@mail.ru (S.K.); oncohematologist@mail.ru (H.J.); parovichnikova.e@blood.ru (E.P.); 2School of Medicine, Lomonosov Moscow State University, 27-1, Lomonosovsky Prospect, 119991 Moscow, Russia; kozh.yana@mail.ru

**Keywords:** short tandem repeat (STR), acute lymphoblastic leukemia, loss of heterozygosity (LOH), uniparental disomy (UPD), chromosomal microarray (CMA)

## Abstract

Despite the introduction of new technologies in molecular diagnostics, one should not underestimate the traditional routine methods for studying tumor DNA. Here we present the evidence that short tandem repeat (STR) profiling of tumor DNA relative to DNA from healthy cells might identify chromosomal aberrations affecting therapy outcome. Tumor STR profiles of 87 adult patients with de novo Ph-negative ALL (40 B-ALL, 43 T-ALL, 4 mixed phenotype acute leukemia (MPAL)) treated according to the “RALL-2016” regimen were analyzed. DNA of tumor cells was isolated from patient bone marrow samples taken at diagnosis. Control DNA samples were taken from the buccal swab or the blood of patients in complete remission. Overall survival (OS) analysis was used to assess the independent impact of the LOH as a risk factor. Of the 87 patients, 21 were found with LOH in various STR loci (24%). For B-ALL patients, LOH (except 12p LOH) was an independent risk factor (OS hazard ratio 3.89, log-rank *p*-value 0.0395). In contrast, for T-ALL patients, the OS hazard ratio was 0.59 (log-rank *p*-value 0.62). LOH in particular STR loci measured at the onset of the disease could be used as a prognostic factor for poor outcome in B-ALL, but not in T-ALL.

## 1. Introduction

Acute lymphoblastic leukemia is a malignant neoplasm characterized by the proliferation of lymphoid cell precursors leading to infiltration of the bone marrow by lymphoblasts. Risk stratification is based on demographic and clinical factors (such as age, ethnicity, gender, blood count at diagnosis, cell lineage, CNS involvement), genetic features of tumor cells, and response to treatment [1]. Conventional cytogenetic analysis complemented with fluorescence in situ hybridization (FISH) is used to detect chromosomal abnormalities and determine risk groups. However, due to inadequate specimens or lack of mitotic cells, these methods are often ineffective. Among ALL patients with a cytogenetic result, 15% to 50% have a normal karyotype [2,3]. Moreover, the above methods are unable to detect copy number neutral loss of heterozygosity (LOH), which can result in deregulation of cell division and apoptosis through the deletion of tumor suppressor genes [4]. Copy number neutral LOH in malignant tumors is a result of somatic uniparental disomy (UPD). UPD occurs due to mitotic homologous recombination events, as attempts to correct the deletion of chromosomal material by using the remaining alleles as a template, or mitotic errors including chromosomal missegregation [5]. Thus, LOH analysis can detect areas of allelic loss not revealed by standard cytogenetic analysis. Moreover, LOH analysis is an effective tool for identifying “masked hypodiploidy”, when patients do not have a karyotypically visible clone with ≤43 chromosomes and, instead, their abnormal karyotypes contain 50–78 or more chromosomes from doubling of previously hypodiploid cells [6]. Genetic instability of tumor cells can lead to a change in the profile of short tandem repeats (STRs, or microsatellites) which are hypervariable and highly mutable simple repetitive DNA sequences [7]. Several studies regarding LOH analysis of childhood ALL using microsatellite markers have been published [8,9,10,11,12]. In these studies, the highest rates of allelic losses were observed in chromosomes 9p and 12p, and a lower frequency of LOH was observed at 5p, 6q, 10p, and 20q. Takeuchi et al. also evaluated the correlation between LOH analysis findings and clinical characteristics and showed that those with 6q LOH had lower WBC counts, patients with 9p LOH more frequently had CNS involvement, individuals with 11q LOH had a good response to induction chemotherapy, and those with 12p LOH had a higher event-free survival rate [8]. There were few genome-wide LOH studies using microsatellite markers of adult ALL [13]; several studies of adult ALL patients were dedicated to LOH in individual chromosomes [14,15,16,17]. In all studies, the frequency of LOH detected by microsatellite analysis was much higher than the reported frequency of cytogenetic aberrations on corresponding chromosomal arms. Our objective was to identify LOH in the blast cells of the patients with B- and T-ALL at diagnosis and to analyze therapy outcomes according to LOH status.

## 2. Materials and Methods

Samples from 87 patients with newly diagnosed nontreated Ph-negative ALL (40 B-ALL, 43 T-ALL, 4 MPAL) were included in the study. The median age was 33 (18–55 years), 52 (60%) were men, and 35 (40%) were women. All patients were treated according to the “RALL-2016” regimen (ClinicalTrials.gov identifier: NCT03462095) at the National Research Center for Hematology (Moscow, Russia). Cryopreserved pretreatment bone marrow samples and the results of the standard cytogenetic analysis were available for these patients.

Tumor DNA was isolated from patients’ bone marrow samples taken at diagnosis. Control DNA was isolated from the buccal swab or blood of patients in complete remission.

STR profiles for each pair of samples (tumor/control) were assessed by PCR with primers to 19 STR loci and amelogenin locus available in COrDIS Plus multiplex kit (Gordiz Ltd., Moscow, Russia). We used the following markers: D1S1656 (locus 1q42), D2S441 (locus 2p14), D3S1358 (locus 3p21.31), D5S818 (locus 5q23.2), D7S820 (locus 7q21.11), D8S1179 (locus 8q24.13), D10S1248 (locus 10q26.3), D12S391 (locus 12p13.2), D13S317 (locus 13q31.1), D16S539 (locus 16q24.1), D18S51 (locus 18q21.33), D21S11 (locus 21q21.1), D22S1045 (locus 22q12.3), CSF1PO (locus 5q33.1), FGA (locus 4q31.3), SE33 (locus 6q14), TH01 (locus 11p15.5), TPOX (locus 2p25.3), VWA (locus 12p13.31), amelogenin X (locus Xp22.1–22.3), and amelogenin Y (locus Yp11.2). For fragment analysis of PCR products, genetic analyzer ABI 3130 (Thermo Fisher Scientific, USA) was used. STR profiles were then analyzed using GeneMapper Software. LOH status was not detected for patients with blast percentage in bone marrow less than 50% (four B-ALL patients and one T-ALL patient).

For all patients with LOH in STR loci, tumor DNA lesions were verified by chromosomal microarray (CMA CytoScan HD) (Figure 1).

Statistical analysis: We used standard methods of descriptive statistics, frequency, and event analysis. To test hypotheses about the differences in the distributions of categorical features in the comparison groups, the analysis of contingency tables was used. The χ^2^ criterion was used to assess the significance; odds ratio (OR) with the corresponding 95% confidence interval (CI) was used as a measure of connection. To test the hypotheses about the presence of differences in the distributions of numerical indicators in the comparison groups, the nonparametric Mann–Whitney rank test was used. In event analysis, the Kaplan–Meier estimates were used to assess the distributions, and the log-rank test was used to assess the statistical significance of differences in the groups. Statistical analysis was performed using the procedures of the SAS 9.4 package. Censoring was done at the date of the last contact for surviving subjects. EFS was defined as the interval from complete remission to relapse or death or last contact. OS was defined as the interval from randomization to death or last contact.

## 3. Results

The main characteristics of patients by phenotype are included in Table 1. Among 87 patients, 37 had normal cytogenetic karyotype, 45 had abnormal cytogenetic karyotype, and in 5 patients it was not possible to determine the karyotype by cytogenetic analysis due to the absence of mitosis in the test material. After comparing the STR profiles of the DNA of tumor cells and healthy cells for each patient, we found a loss or decrease in the signal from one of a pair of alleles at heterozygous STR loci in 21 patients out of 87 (24.1%). Of these 21 patients, 7 had a normal tumor karyotype and 14 had an abnormal one. In 8 out of 14 patients with an abnormal karyotype, LOH was also found on chromosomes that were not aberrant according to cytogenetic analysis data. Thus, analysis of STR profiles revealed additional abnormalities in seven patients with normal karyotype and eight with abnormal karyotype, which is 17.2% of the cohort.

The most frequent LOH at 12p was detected in eight patients (9.2% of patients). Other frequent loci with LOH included 6q (five patients, 5.7%), 1q (four patients, 4.6%), and 4q (four patients, 4.6%). It is noteworthy that in T-ALL the loss of heterozygosity covers a smaller set of STR loci than in B-ALL. The full distribution of LOH by loci is presented in Appendix A.

According to OS analysis, LOH 12p seems to be a marker of a favorable prognosis (Figure 2). Patients with the 12p LOH are excluded from the risk group because 12p LOH is not associated with worse outcome according to our data and [8,18].

We have checked the distributions of all major risk factors such as age, gender, leukocyte count at diagnosis, LDH level, blast cell percentage in the bone marrow and peripheral blood, immunophenotype, standard cytogenetic karyotype, CNS leukemia, extramedullary disease, and complete remission/refractory after second induction (+70 days) in groups B-ALL and T-ALL in relation to LOH status. We have found no significant differences between the analyzed groups (Table 2 and Table 3), with one exception: in T-ALL the percentage of blast cells in bone marrow was significantly higher in the LOH-positive group (*p* = 0.02).

Therefore, we conclude that the influence of LOH on clinical output is not due to its correlation with some known risk factors. For B-ALL patients, LOH (excepting 12p LOH) was an independent risk factor (OS hazard ratio 3.89, log-rank *p*-value 0.0395) (Figure 3a). For T-ALL patients, OS hazard ratio was 0.59 (log-rank *p*-value 0.62) (Figure 3b).

The main findings concerning karyotypes and microarray data of the patients with aberrant tumor STR profiles are summarized in Table 4.

## 4. Discussion

It is known from classical studies that karyotyping of bone marrow before treatment is an important part of adult acute lymphocytic leukemia diagnosis and provides important prognostic information [19]. Therefore, standard cytogenetic analysis has been routinely performed in all patients at the onset of the disease. With the development of molecular technologies, new methods for assessing the tumor genome have been proposed, many of which are used in scientific works, but have not been introduced into routine practice yet. With the development of microarray analysis, it has become possible to screen the genome for copy number alterations (CNAs) and regions of copy neutral loss of heterozygosity (cnLOH) that are not detectable by G-banding or fluorescence in situ hybridization (FISH) [20,21].

In this study, we aimed to develop a robust and simple molecular method to identify patients with additional chromosomal lesions otherwise detectable by CMA only. Therefore, we chose STR-PCR to compare tumor and “healthy” genomic DNA. Due to the high polymorphism of STR loci in the population, most of them are present in humans in a heterozygous form; therefore, a wide STR panel could reveal the loss of heterozygosity, which later might be verified by CMA.

We performed LOH analysis of chromosome arms 1q, 2p, 3p, 4q, 5q, 6q, 7q, 8q, 10q, 11p, 12p, 13q, 16q, 18q, 21q, and 22q using STR markers. LOH was detected in 24.1% of ALL patients, with the highest frequency of LOH in the 12p chromosome arm. As compared with standard cytogenetic analysis, we were able to detect additional abnormalities in 15 (17.2%) out of 87 patients using STR profile analysis. Thus, cytogenetic analysis is insufficient for the detection of LOH in ALL patients.

We observed a statistically significant association of poor outcome with the LOH in STR loci (excluding 12p LOH) measured at the onset of B-ALL, but not of T-ALL. For B-ALL patients, the most abundant STR loci with LOH observed were D1S1656 (1q42, four patients), SE33 (6q14, four patients), and FGA (4q31.3, three patients). No significant associations of LOH in particular loci with clinical events were found. Patients 29, 32, 63, and 89 did not achieve remission and died; patients 39, 82, and 91 are still on therapy; and patients 45 and 71 are in remission. One can speculate that gains might be less harmful (patients 45 and 71, Table 4) since no loss of genetic material occurred. One limitation of our study is the fact that our STR panel did not contain markers for several chromosome arms, including 9p. For one T-ALL patient with normal karyotype and the only one LOH in 10q26.3 (patient 2), additional cnLOH at 9p (24.3–13.3) was revealed by chromosomal microarray. CMA also showed the microdeletion at 9p21.3 (21656682_22304230) affecting the region of cnLOH (Appendix A). As a result of these two chromosomal aberrations leading to a homozygous form of deletion, a cluster of MTAP, CDKN2A-AS1, CDKN2A, CDKN2B-AS1, and CDKN2B genes containing three oncosuppressor genes involved in the regulation of antiproliferative and proapoptotic activities of Rb1 and p53 was completely deleted from the tumor cell genome. The affected patient had the highest level of leukocytes (834 × 109/L) at diagnosis; he died from refractory disease. Abnormalities affecting the 9p chromosome are frequent in ALL patients and associated with a negative impact on survival and increased risk of treatment failure [22,23,24]. Therefore, the STR panel used in our study needs to be further expanded with markers located in the most frequent and prognostically important loci.

We also tried to assess the LOH 12p as a separate risk factor. LOH 12p seems to be a marker of a favorable prognosis, but the group of patients with LOH 12p without any other LOH variants is too small (four patients only) to assess statistical significance.

The predictive value of copy number variations in acute B-cell lymphoblastic leukemia in children is noted in the work of Rosales-Rodríguez et al. [25]. CNA risk classifiers have become effective tools for predicting disease recurrence. However, their clinical use has yet to be translated into routine practice.

Detecting CNA and mutations simultaneously during diagnosis and relapse is clinically important for predicting tumor clones’ evolution in B-ALL patients. New genomic strategies for detecting different genetic lesions are being addressed by developing specific panels for each type of hematologic malignancy aimed to facilitate risk stratification and improve customized treatment strategies for ALL [26].

A recent study proposes an additional stratification index for B-ALL patients based on changes in the DNA copy number variations identified by multiple ligation probe amplification (MLPA) [27]. Copy number alterations in the aforementioned study, as well as in our study, were associated with poor prognosis and the lowest OS as well as EFS rates [27].

## 5. Conclusions

One can conclude that STR analysis is an effective, easy, and low-cost tool for detecting LOH in ALL patients, although it is limited by incomplete coverage by common STR panels. In contrast, CMA with single nucleotide polymorphism (SNP) probes has more extensive coverage and has proven its effectiveness in identifying both copy number alterations and copy neutral LOH [28,29,30,31,32]. Thus, CMA can provide clinically significant information that complements cytogenetic analysis. At the same time, translation of CMA into routine clinical testing is hampered by its comparatively complicated use and high costs. Here, STR in our view represents a viable alternative. We therefore in conclusion propose applying CMA to study LOH and thereby identify the whole picture of clinically relevant loci. Based thereon, extended STR panels can be developed to be readily translated into routine use in ALL diagnostics.

## Figures and Tables

**Figure 1 genes-13-00398-f001:**
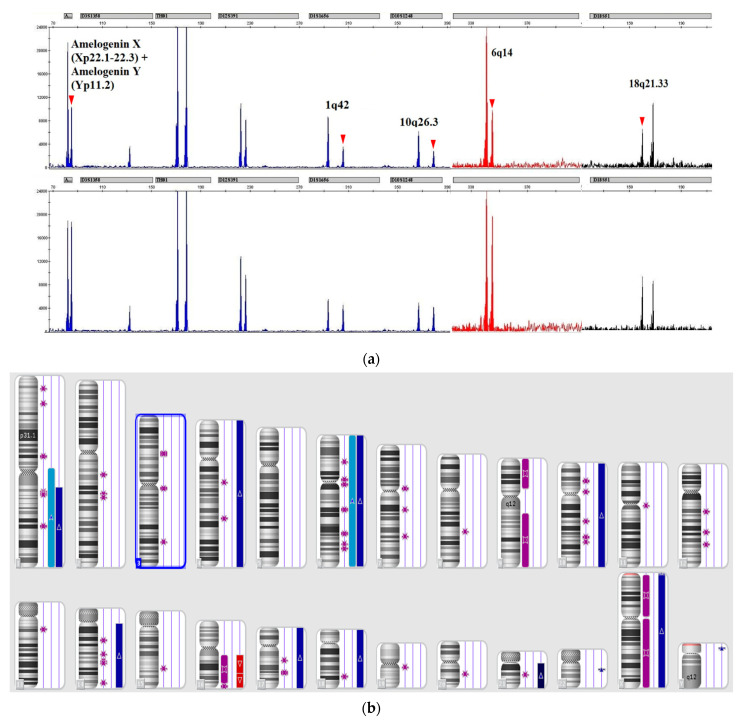
LOH patterns in STR profile of B-ALL patient 45 (**a**,**b**) molecular karyoview of CMA. The karyoview presents a diagram of all copy number variations (red bar for loss; blue bar for gain) and copy neutral LOH (purple bar). (LOH at 1q42, 10q26.3, 6q14, and 18q21.33 and disbalance at amelogenin X/amelogenin Y were verified by CMA as gains in areas including aberrant STR loci).

**Figure 2 genes-13-00398-f002:**
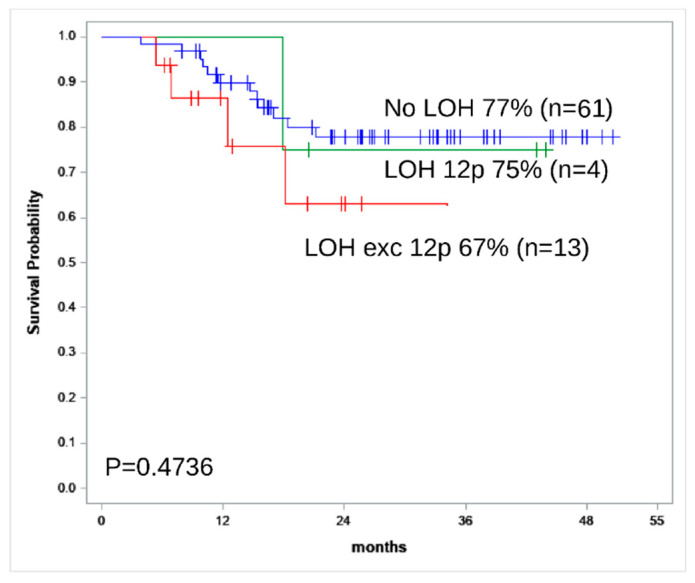
Kaplan–Meier survival curve for OS estimates according to the LOH status in tumor STR profile (LOH (except LOH 12p), red line; LOH 12p, green line; no LOH, blue line). Four patients with LOH 12p and additional LOH in several loci and five patients with unidentified LOH status were excluded from analysis.

**Figure 3 genes-13-00398-f003:**
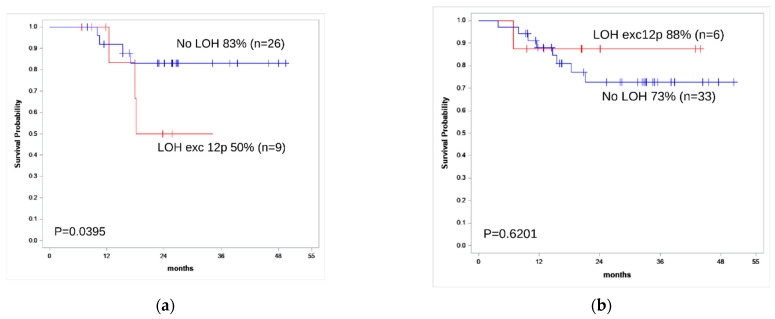
Kaplan–Meier survival curve for OS estimates according to the LOH status in tumor STR profile for B-ALL (**a**) and T-ALL (**b**) patients. Patients 3, 4, 43, and 65 were excluded from the LOH group due to 12p LOH.

**Table 1 genes-13-00398-t001:** The main characteristics of the patients.

*	B-ALL (*n* = 40)	T-ALL (*n* = 43)	MPAL (*n* = 4)
Male:Female	20:20	28:15	4:0
Age, median	36 (19–61) years	33 (20–50) years	32 (27–36) years
Leukocytes, 10*9/L	9.25 (1.39–593.48)	44.9 (1.09–833.94)	3.82 (1.79–5.69)
LDH	964 (221–20,062)	1447(120–20,064)	308 (154–1052.6)
Blast cells in peripheral blood, %	56 (0–98)	67.5 (0–96)	1 (0–71)
Blast cells in the bone marrow, %	80 (0–96.8)	86.6 (0–100)	36.8 (31.2–74.8)
Immunophenotype, EGIL, WHO	B-I 7 (17.5%)	ETP (T-I) 4 (9.3%)	T/myelo 2 (50%)
	B-II 31 (77.5%)	ETP (T-II) 2 (4.7%)	B/myelo 1 (25%)
	B-III 1 (2.5%)	T-II 11 (25%)	T/B 1 (25%)
	B-IV 1 (2.5%)	T-III 23 (54%)	
		T-IV 3 (7%)	
Standard cytogenetics			
+ mitosis	37 (92.5%)	41 (95.3%)	4 (100%)
− mitosis	3 (7.5%)	2 (4.7%)	
Karyotype			
Normal	16 (40%)	17 (39.5%)	4 (100%)
Abnormal:	21 (52.5%)	24 (55.8%)	
Complex (>3 chromosomal aberrations)	6	5	
Hyperploid	4	1	
Other	11	17	
CNS leukemia	6 (15%)	9 (20.9%)	0 (0%)
Extramedullary disease	11 (27.5%)	27 (62.8%)	3 (75%)
CR			
After 2nd induction (+70 day)	35	40	2 (50%)
Refractory disease	4	3	2 (50%)
Death in CR	1		

* B-ALL, B-cell lymphoblastic leukemia; T-ALL, T-cell lymphoblastic leukemia; MPAL, mixed phenotype acute leukemia; LDH, lactate dehydrogenase; EGIL, European Group on Immunological Classification of Leukemia; WHO, World Health Organization; ETP, early T-cell precursor; CNS, central nervous system; CR, complete remission.

**Table 2 genes-13-00398-t002:** The main characteristics of B-ALL patients according to the LOH status. *p*-value from χ^2^ was used to estimate significance for the tables of 2 × 2.

	LOH-Positive * (*n* = 10)	LOH-Negative (*n* = 26)	*p* χ^2^
Male:Female	6:4	11:15	
Age, median	34.5 (19–61) years	36 (21–53) years	
Leukocytes, 10*9/L	10.76 (1.40–34.39)	5.66 (1.09–593.48)	1.000
LDH, U/L	1439 (221.00–20,062.00)	615.50 (293–7348.8)	0.260
Blast cells in peripheral blood, %	52.5 (1–89)	15 (0–98)	0.540
Blast cells in the bone marrow, %	84.20 (70–95.8)	79 (0–200)	0.300
Immunophenotype, EGIL, WHO	B-II 9 (90%)	B-I 7 (27%)	0.161
	B-IV 1 (10%)	B-II 19 (73%)	
Standard cytogenetics			
+ mitosis	9 (90%)	24 (92.3%)	0.823
− mitosis	1 (10%)	2 (7.7%)	
Karyotype			
Normal	2 (20%)	10 (38.5%)	0.302
Abnormal:	7 (70%)	14 (53.8%)	
Complex (>3 chromosomal aberrations)	5 (3 deaths)	1	
Hyperploid	1	3 live	
Other	1	10	
CNS leukemia	2	3	0.511
Extramedullary disease	2	8	0.519
CR			
After 2nd induction (+70 day)	10	23	0.262
Refractory disease	0	3	
2-year OS	50%	83%	0.040
2-year RFS	50%	85%	0.030

* For four B-ALL patients, LOH status was not detectable due to blast percentage in bone marrow being less than 50%.

**Table 3 genes-13-00398-t003:** The main characteristics of T-ALL patients according to the LOH status. *p*-value from χ^2^ was used to estimate significance for the tables of 2 × 2.

	LOH-Positive * (*n* = 9)	LOH-Negative * (*n* = 33)	*p* χ^2^
Male:Female	6:3	21:12	
Age, median	31.5 (20–37) years	33 (20–50) years	
Leukocytes, 10*9/L median	80.35 (9–833.94)	23 (2.06–333.21)	0.210
LDH, U/L	1738.5 (415–20,064.3)	1299 (120–7633.2)	0.300
Blast cells in peripheral blood, % median	75.5 (6–96)	60.0 (0–92)	0.160
Blast cells in the bone marrow, % median	93.4 (66–98)	82.4 (0–100)	0.020
Immunophenotype, EGIL, WHO	T-II 1 (11%)T-III 8 (89%)	ETP (T-I) 4 (12%) ETP (T-II) 2 (6.1%) T-II 9 (27.3%) T-III 15 (45.5%) T-IV 3 (9.1%)	
Standard cytogenetics			
+ mitosis	9 (100%)	31 (94%)	0.450
− mitosis	0	2 (6%)	
Karyotype			
Normal	3 (33.3%)	12 (36.4%)	0.770
Abnormal:	6 (66.7%)	19 (57.6%)	
Complex (>3 chromosomal aberrations)	-	5	
Hyperploid	-	1	
Other	6	13	
CNS leukemia	2 (22.2%)	6 (18%)	0.785
Extramedullary disease	5 (55.6%)	24 (72.7%)	0.324
CR			
After 2nd induction	8	31	
Refractory disease	1	2	0.603
2-year OS	88%	73%	0.62
2-year RFS	100%	69%	0.16

* For one T-ALL patient, LOH status was not detectable due to blast percentage in bone marrow being less than 50%.

**Table 4 genes-13-00398-t004:** The karyotypes and microarray data of the patients with aberrant STR profiles in tumor DNA.

##	All Phenotype	Aberrant STR Loci	Karyotype	Microarray
29	B-II	1q42, 13q31.1, 6q14	45,XX,del(6)(q12),i(7)(q10),-13,add(19)(p?;q?)[10]/46,XX,i(7)(q10),del(13)(q22), add(19)(p?;q?)[1]/46,XX,add(3)(q26),del(6)(q12),i(7)(q10),add(19)(p?;q?)[1]/46,XX[8]	Gain 1q, loss 6q, loss 7p, gain 7q, loss 13q
32	B-II	3p21.3	51,XX,del(1)(p31),+1,del(4)(q28),+5,+8,der(9),del(13)(q14q22),+13,der(15),del(17)(q23),+21[15]/46,XX[15]	Gain 1q, LOH 3 chr, gain 5 chr, 7 chr, 8 chr, loss 13q, gain 21q
39	B-II	7q21, 4q31, 16q24.1, 5q33, 5q23.2	46,XY[27]/45,X,-Y[1]/46,XY,del(1)(q31)[1]/46,XY,?add(1)(q44)[1]	Gain 1 chr, 2 chr, loss 5q, gain 6 chr, loss 9chr, gain 10 chr, 12 chr, 14q, loss 15q, 16q, gain 18chr, 19 chr, 21q, 22q
45	B-II	Imbalance X/Y, 1q42, 10q26, 6q14, 4q31,18q21.33	n/a	Gain 1q, 4 chr, 6 chr, LOH 9 chr, gain 10 chr, 14q, loss16q, gain 17 chr, 18 chr, 21 chr, X chr
63	B-II	5q23.2	46,XX[20]	n/a
65	B-II	12p13.2	46,XY,-7,add(12)(p?),add(12)(p?),-17,+2mar[cp8]/46,XY[22]	Loss 7q, 12p, gain 17p
89	B-IV	1q42, 12p13.2, 16q24.1, 22q12.3	49,XY,+1,del(1)(p10),der(3),del(6)(p23),der(8),del(9)(p23),+12, t(14;18)(q32;q21),+16,der(17)[20]	Gain 1q, LOH 2p, 2q, 3p, 3q, gain 4p, LOH 5p, 5q, gain 8q, 10p, LOH 12p,16p, 16q, gain 18q
71	B-II	Imbalance X/Y, 8q24.13, 10q26.3,18q21.33, 4q31.3, 6q14	57,XY,+X,+4,+6,+8,+10,+14,+17,+18,+21x2,+mar[14]	Gain 4p, 4q, 6p, 6q, 8p, 8q,10p, 10q, 14q,16q, 17p, 17q, 18p, 18q,20q, Xp
82	B-II	6q14	46,XX,?del(6)(q22),-7?,add(19)(p?q?) or der(19)t(1;19)(q23;p13), +mar,inc[cp7]/46,XX[3]	Gain 1q, loss 6q, 9p, gain 9q
91	B-II	1q42, 5q23.2	46,XY,der(7)t(1;7)(q10;p21)[15]/46,XY,der(7)t(1;7)(q10;p21)x2[1]/46,XY[14]	Gain 1q, LOH 5q23
2	T-III	10q26.3	46, XY[30]	LOH 9p, LOH 10q
3	T-II	12p13.2, 12p13.3	46, XY	Gain 5p, 6q, loss 9p, 12p, gain 17q, 18p, 18q, 19p
4	T-III	12p13.2	47,Y,?t(X;19)(p21;?p13?q13),der(18),+der(19)t(X;19)(p21;?p13?q13) or der(19)[11]/46,XY[3]	Loss 12p, 18p, 18q, gain 19p, Xp
33	T-III	12p13.2, 12p13.31, 4q31.3	46,XY,t(7;9)(p14;p23),der(14)[26]/46,XY[7]	LOH 4q, loss 7p, LOH 9p, gain 9q, LOH 12p, loss 14q
43	T-III	12p13.2	No data	Loss 9p, gain 9q, loss 12p,12q, 13q
64	T-III	21q21.1, EMAST 4q31.3	46,XX[28]/47,XX,+mar[1]/47,XX,+9,i(9)(q10)[1]	Gain 21q
66	T-III	10q26.3	46,XY[19]/46,XY,del(7)(p11)[1]	n/a
85	T-III	11p15.5	46,XY[20]	n/a
88	T-III	6q14	t(2;14)(q36;q22)[15]/46,XX[10]	Loss 3p, gain 3q, 5p, loss 6q
7	MPAL	5q33.1	46,XY,add(1)(q44), der (17) i ?(17)(q10)[1]/46,XY[24]	Gain 3q, loss 5q, loss 12q, LOH 17p
99	MPAL	2p14, 7q21, 13q31, 16q24, 18q21, 21q21, 2p25, 12p13.31	n/a	LOH 8q, loss/LOH 2p, LOH 7q21.3q22.1 and many mozaic LOH less than 5 × 10^6^ bp

## patient number.

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
