# Peer review of "Loss of Heterozygosity in the Tumor DNA of De Novo Diagnosed Patients Is Associated with Poor Outcome for B-ALL but Not for T-ALL"

_genes, 2022, doi:10.3390/genes13030398_

Round 1

Reviewer 1 Report

In the present manuscript Risinskaya et al., demonstrated that analysis LOH at STR loci can be used as a prognostic factor to determine the clinical outcome in B-ALL but not in T-ALL patients. Overall, this study is adequately designed and the results are well interpreted. Following are my minor concerns:

  1. I found many typographic errors e.g in the title "Patientis", Line no 141 "LHDlevel" line no 160 "12pLOH.
  2. The size and resolution of both images in Fig.1 a&b are below acceptable level. It would be good to upload large images with high resolution and proper labeling. It's hard to find which image belongs to which patient category.
  3. In B-ALL patient group LOH exp 12p (n=9), what are the most abundant STR loci with LOH you observed? Discuss with clinical correlation.

Reviewer 2 Report

To the authors:

This study aims to identify Loss of Heterocygosity (LOH) as a method of diagnosis of blast cells from patients with B and T-ALL. Furthermore, this study seeks to analyze therapy outcomes according to LOH status. In general, ALL is  one of the most common type of cancer and leukemia in children, that progresses rapidly without treatment.  Therefore, studies such as the present one could provide valuable information on how to better detect the disease and develop new strategies to improve the survival of ALL patients. However, despite the significant amount of work done, some weaknesses need to be addressed before publication.

General Comments:

In general, the manuscript is well written and reads easily. The experiments appear to be sound, thoroughly well and planned and carried out.

However, the authors emphasize that the main objective of the paper is to demonstrate that STR-based karyotyping and chromosomal microarray are two useful techniques for better diagnosis of ALL. They show in Figure 1 a STR profile and molecular karyoview of patient 45#. Why did the authors decided to show this patient specifically? In the same way, this figure should be bigger in orther to facilitate its study.

They do not show any figures or supplemental material showing such karyotyping and microarrays of the others patients. I suggest the authors to make a general figure showing the main findings of the karyotypes and microarrays performed on the patients, and not just describe them in the results section. In the same way, I suggest to the authors prepare all the raw data as a supplemental material.

Finally, I would like to congratulate the authors for including a limitation of their study.

Round 2

Reviewer 2 Report

Comments to the Author:

I commend the effort made by the authors to revise this manuscript. The authors have answered correctly all my questions and they have added more data into the figures as I suggested to them. The authors have improved the size of Figure 1 as I requested them. In the same way, they have added one table more to show the karyotypes and microarray data of all the patients, including a supplemental table.  Overall, I consider that now the manuscript is more solid and it has improved considerably for its publication. For this reason, I encourage to editor to consider this manuscript for publication for the interesting value of the study realized, that now it is a much more robust study.
